# Latest Evidence on Post-Prostatectomy Urinary Incontinence

**DOI:** 10.3390/jcm12031190

**Published:** 2023-02-02

**Authors:** Mauro Gacci, Cosimo De Nunzio, Vasileios Sakalis, Malte Rieken, Jean-Nicolas Cornu, Stavros Gravas

**Affiliations:** 1Department of Minimally Invasive and Robotic Urologic Surgery and Kidney Transplantation, Careggi University Hospital, Largo Brambilla 3, 50134 Florence, Italy; 2Department of Experimental and Clinical Biomedical Science, University of Florence, 50134 Florence, Italy; 3Department of Urology, Sapienza University, Ospedale Sant’Andrea, 00185 Rome, Italy; 4‘Agios Pavlos’ General Hospital of Thessaloniki, 55134 Thessaloniki, Greece; 5Alta Uro AG, 4051 Basel, Switzerland; 6Faculty of Medicine, University of Basel, 4056 Basel, Switzerland; 7Urology Department, CHU de Rouen, F-76000 Rouen, France; 8Medical School, University of Cyprus, Nicosia 2029, Cyprus

**Keywords:** prostate cancer, radical prostatectomy, robotic prostatectomy, urinary incontinence, post prostatectomy incontinence, PPI, voiding diary, pad test, antimuscarinics, beta-3 agonists, duloxetine trans-obturator slings, artificial urinary sphincter

## Abstract

A radical prostatectomy is frequently used as the first-line treatment for men with prostate cancer. Persistent urinary incontinence after surgery is one of the most severe adverse events. We report the results of a comprehensive literature search focused on post-prostatectomy urinary incontinence (PPI), performed by a panel of experts on non-neurogenic lower urinary tract symptoms. The data on the prevalence and timing of PPI are very heterogeneous. The etiology of PPI can be multifactorial and mainly dependent on patient characteristics, lower urinary tract function or surgical issues. The medical history with a physical examination, the use of validated questionnaires with a voiding diary and pad tests are determinants in identifying the contributing factors and choosing the right treatment. Lifestyle intervention and urinary containment are the most frequently used strategies for the conservative management of PPI, while antimuscarinics, beta-3 agonists and duloxetine (off-label) are drugs indicated to manage PPI with a concomitant overactive bladder. Surgical therapies for the management of post-prostatectomy SUI include non-adjustable trans-obturator slings in men with mild-to-moderate incontinence and an artificial urinary sphincter in men with moderate-to-severe incontinence.

## 1. Introduction

Prostate cancer is the most common malignancy in elderly men [1]. A radical prostatectomy is extensively used as a first-line of treatment of clinically localized prostate cancer. Many men report a transitory worsening of their urinary function immediately after surgery, with a subsequent progressive recovery over 1 to 2 years [2,3].

However, there is a cohort of patients who suffer from persistent incontinence, with a severe decline in quality of life (QoL), causing physical, emotional, social, occupational and hygienic problems [3]. In addition, incontinence carries a significant cost for both patients and health systems [3]. 

The dissimilar definitions of continence (self-reported by patients [4] or measured objectively by a physician) and postoperative timing (the “early postoperative” period reportedly ranges from 1 week to 3 months) can have significant effects on the overall rate and timing of continence recovery 3 and 6 months postoperatively [5]. Moreover, the data from the current literature are available only from inconsistent and contradictory single-center prospectives or retrospective studies.

As a result, there are significant gaps in our knowledge, due to the high variability in the reported rates of post-prostatectomy incontinence (PPI), the fact that the pathophysiological mechanism is not fully understood, the lack of a universal consensus on the necessary diagnostic work-up and the optimal and timely selection of the appropriate treatment.

The following manuscript aims to report a practical, evidence-based review of the epidemiology, risk factor pathophysiology, diagnostic work-up and treatment options for men with PPI, in order to assist clinicians in identifying which patients will or will not experience incontinence recovery and set realistic expectations regarding the short-, medium- and long-term efficacy of available therapeutic options).

## 2. Materials and Methods

The authors of the present review have been selected from the European Association of Urology (EAU) guidelines panels on non-neurogenic male LUTS [6]. A comprehensive literature search, limited to the best available evidence and studies published in the English language, on male urinary incontinence, including the data on PPI, was performed [7]. From this literature review, only manuscripts focused on PPI were selected. 

The full details of the search strategy are available on the following website: https://d56bochluxqnz.cloudfront.net/documents/guideline-appendices/management-of-non-neurogenic-male-luts/Search-strategies-Non-Neurogenic-Male-LUTS-guideline-2022.pdf (accessed on 8 February 2022). In particular, from a total of 1054 unique records, 663 were excluded (due to: being female, animal, child, duplicates, protocols, abstracts or papers on neurogenic UI). From the remaining 391 papers on male UI evaluated by our panel, 154 were included in the present review.

Further information is available on a specific URO webinar from the European School of Urology (ESU), reporting the surgical anatomy, pathophysiology and both medical and surgical treatments for PPI [8].

## 3. Results

### 3.1. Epidemiology and Risk Factors 

The prevalence of PPI varies from 1% to 87%, depending on the definition, timing of evaluation, surgical approach and who carries out the assessment [9,10,11]. Most patients experience transitory incontinence immediately after RP, reaching a grade of complete continence within 2–3 months [12]. Several studies report a progressive return of continence up to one year after RP, with a continence rate ranging from 68 to 97% at 12 months [13,14,15,16,17], while a progressive further improvement could be registered up to 2 years [18]. 

PPI is a multi-factorial condition that can be due to intrinsic sphincter deficiency (ISD) with or without pre-existing or arising de novo post-RP bladder dysfunction [19,20,21]. The majority of evidence in the literature supports that the most common contributing factor to PPI is ISD, with an incidence ranging from 67 to 92.4% [22,23,24]. ISD is mainly due to the manipulation of the rhabdosphincter during an apical dissection of the prostate [19]. Bladder dysfunction, such as detrusor overactive (DO) and detrusor underactive (DU), and impaired bladder compliance can be present in 33–61% of patients but is the solitary cause of incontinence in only 1.5–4%. When present in association with ISD, bladder dysfunction may not always be clinically relevant [19,24,25,26,27,28]. Bladder dysfunction can arise de novo after prostatectomy, and a series of causes have been supposed: partial somatic and autonomic decentralization resulting from bladder mobilization [29,30], as well as inflammation/infection or geometric bladder wall alteration associated with pre-existing hypoxemia [31]. Post-prostatectomy anastomotic stricture and subsequent obstruction may also play an important role in the development of de novo DO [32]. However, bladder dysfunction may have been present preoperatively, for example, as a condition of DO secondary to pre-existing bladder outlet obstruction (BOO). Nevertheless, a history of bladder neck contracture can be a risk factor for PPI. 

Age is a well-known predictive factor for PPI, as it plays an important role in the recovery of continence but, also, in pre-existing urinary continence [33,34,35,36,37,38]. Patients with diabetes mellitus need a longer time to regain continence, as it has been reported to be a significant disadvantage in gaining urinary continence compared to non-diabetics in the first 18 months after RP, but not in the total long-term continence rate [39,40]. The correlation of other risk factors of PPI, such as BMI, metabolic syndrome and prostate volume, is still controversial [37,41,42,43,44,45,46,47,48,49]. Anatomic characteristics, including the thickness of the pelvic diaphragm on sagittal imaging, the ratio of the levator ani on the axial image to the prostate volume and the functional urethral and sphincter length, have been attracting increasing attention as factors that might be mitigated with customized surgical techniques [50,51]. In addition, regarding surgery-related risk factors, it has been demonstrated that the surgeon’s experience and surgical technique are important determinants of post-operative incontinence rates [26,52,53,54,55]. 

### 3.2. Pathophysiology

Historically, intraoperative damage to the urethral sphincter or its innervation was supposed to be the main cause of incontinence after radical prostatectomy [56]. However, the true etiology has not been completely understood, and currently it is thought to be multifactorial: the factors affecting post-surgery continence can be classified as patient/biologically/surgically related. 

#### 3.2.1. Patients’ Related Factors

Age can have an adverse impact on continence recovery after surgery, considering that elderly patients present other possible risk factors, such as larger prostates, a higher frequency of concomitant overactive bladder or comorbidities when compared to younger patients. Mandel P et al. [57], in a study based on more than 8000 patients, found that the 1-year continence rate decreases in accordance with the patient’s age, from 93.2% in <65 years to 86.5% in ≥75-year-old patients. Likewise, a Charlson comorbidity index was an independent predictor of continence rate (OR 1.635, *p* = 0.009) in a cohort of 308 consecutive laparoscopic prostatectomies [41].

Obesity and patient lifestyles have been also investigated as possible risk factors. Particularly, a body mass index (BMI) >30 and lower physical activity have been thought to negatively affect the probability of recovery of urinary continence 1 year after surgery [58,59] It has been also hypothesized that increased physical activity may reduce the risk of incontinence by increasing muscle tone, while obesity may increase the risk of incontinence by placing additional physical strain on the bladder. Moreover, active and not obese patients may also have a better health consciousness, and thus are more likely to comply with physicians’ recommendations for Kegel exercises [60,61,62].

#### 3.2.2. Biological Factors

The role of post-operative incontinence in preoperative lower urinary tract dysfunction (LUTD) has been investigated. Thus, it has been reported that up to 50% of men suffering from post-operative incontinence present pre-operative LUTD. Giannantoni A et al. [63] detected, prior to surgery, decreased bladder compliance, impaired detrusor contractility and detrusor overactivity in 20.4%, 42.8% and 55.1%, respectively, of their study cohort. These bladder dysfunctions can also arise de novo after radical prostatectomy as the result of anatomic changes, ischemia and denervation of the bladder, and they can contribute to the onset of post-operative urinary incontinence. The relief of a possible concomitant bladder outlet obstruction could also explain a low rate of overactive bladder symptoms (less than 5%) after a radical prostatectomy [64]. The negative impact of moderate/severe preoperative lower urinary tract symptoms (LUTS) or a larger prostate on RRP incontinence has also been deeply investigated [64]. In particular, the prostate shape and size can influence both the urethral length and bladder neck dissection for several reasons, including the possible presence of a middle lobe, a greater gap between the bladder and urethra and a major risk of denervation due to less mobility of the large gland [46]. All these conditions are associated with an increased risk of anastomotic leakage, which determines local fibrosis and has a negative effect on the urethral sphincter function. Boczko et al., also found a 6-month continence rate of 97% in patients with prostate sizes less than 75 g vs. 84% in larger prostates (*p* < 0.05) ([65]. Finally, a functional urethral length (external urethral sphincter length) of at least 2.8 cm was initially suggested as a desirable cut-off to guarantee adequate continence [66]. A statistically significant difference in the maximal urethral closure pressure (68.1 vs. 53.1 cm with water) was observed in patients with a longer functional urethral length (27.6 vs. 20.5 mm) [67]. Likewise, studies based on MRI findings confirm that the likelihood of continence recovery significantly increases with a longer preoperative membranous urethral length [68,69] 

#### 3.2.3. Surgical Factors

Several surgical techniques have been evaluated to reduce the risk of post-operative incontinence. Bladder neck preservation improves continence due to a more detailed dissection that could help to preserve the urethral length and continence structure. A meta-analysis based on thirteen trials has recently confirmed that this technique is associated with an improvement of early urinary continence rates (6 mo, OR = 1.66; 95% CI, 1.21–2.27; *p* = 0.001) and long-term urinary continence outcomes (>12 mo, OR = 3.99; 95% CI, 1.94–8.21; *p* = 0.0002) [70]. Likewise, the preservation or reconstruction of membranous urethral supporting structures seems to significantly improve post-operative continence by fixing the urethra in an anatomic position during enhancement of abdominal pressure (the so-called “Rocco stitch”). The Retzius-sparing robot-assisted prostatectomy (RS-RARP), proposed by Galfano et al. in 2010 [71], has been also associated, in a recent meta-analysis, with a better postoperative continence recovery than conventional RARP [72] (Liu et al., 2022). 

### 3.3. Diagnostic Work-Up

A thorough diagnostic approach is essential to evaluating a patient with incontinence (Figure 1) through a systematic method to identify the contributing factors and to tailor the incontinence therapy according to the patient’s needs and preferences.

#### 3.3.1. Medical History and Physical Examination

A careful clinical history is fundamental to the clinical process, and, despite the lack of formal evidence, there is universal agreement that this should be the first step in the assessment of patients with urinary incontinence (UI) [73,74]. The clinical history alone can categorize UI into stress SUI, urge UUI or mixed MUI, and can recognize those who might benefit from a referral to an appropriate specialist (e.g., pelvic diseases, neurological conditions). Evidence from the female population has shown that urodynamic stress incontinence can be correctly identified in primary care from the clinical history alone, with a sensitivity of 92% and specificity of 56% [75]. The history should include questions about the type, timing and severity of the UI, as well as questions about any other associated urinary symptoms. Additional questions about co-morbidities, medications and overall health status are recommended [73].

A physical examination of males with UI should be similar to that of male LUTS patients [73]. In addition to the abdominoperineal and external genitalia examination, it should include a digital rectal examination and basic neurological assessment. A specific test for male UI patients is the cough stress test [76]. The patient is asked to cough in a standing position with a bladder comfortably full. The leak is quantified based on the male stress incontinence grading (MSIG) scale from 0 to 4. The MSIG scale scores 0 when leakage is reported but not demonstrated, 1 when there are delayed drops only, 2 in the case of early drop leakage without a stream, 3 in the case of early drop leakage with a stream and 4 when there is an early persistent stream. During coughing, pressure can be applied to the perineum aiming to stop leakage (bulbar compression positive test).

#### 3.3.2. Questionnaires, Voiding Diaries and Pad Tests

Specific validated questionnaires help to quantify the severity of UI and assess post-treatment changes [73,74,77,78,79]. The evidence, however, that QoL or condition-specific questionnaires have an impact on the outcome of treatment is limited [73]. Voiding diaries are standardized, semi-objective tools to measure symptom severity [80]. They record the volume and frequency of urination episodes, or any additional information such as fluid intake, incontinence episodes, number of pads used, or other parameters that may be of interest, such as the type of activities during leakage. Observational studies have demonstrated a strong correlation between voiding diaries and standard symptom evaluation, as well as a potential therapeutic benefit [81,82,83,84]. Even though there is no consensus regarding diary duration (1-, 3-, 7-day), an extensive duration reduces patient compliance, while too short a duration produces unreliable measurements [85]. A pad test is used to quantify the severity of UI and to monitor the patient’s response to treatment [73]. It can be either 1 h, office-based, with specific exercises, or 1- to 3-days, home-based, with daily activities. There have been previous attempts to grade UI based on pad test severity as mild, moderate and severe [86]. The usefulness of the pad test in predicting the treatment outcome is uncertain; however, early post-operative testing with pad tests may predict future continence in men after prostatectomy [87,88].

#### 3.3.3. Imaging

Ultrasonography, CT and magnetic resonance imaging can help the clinician to understand the anatomical and functional abnormalities that cause UI and correlate it with the treatment outcome. An attempt to quantify post-prostatectomy urethral hypermobility by transperineal ultrasonography using three reference points was previously described [89]. A post-void residual (PVR) measurement using it can distinguish overflow incontinence from other UI types. The PVR can be measured by portable or US devices, as well as by catheterization; several studies, however, have demonstrated that US measurement of PVR is preferable to catheterization [90,91,92,93,94,95].

#### 3.3.4. Urinalysis

A urinalysis is important in the diagnostic approach of a patient with UI, because it may recognize infections, proteinuria, hematuria, or glycosuria, which necessitate further evaluation [96,97,98,99]. Frequently, UI may complicate a symptomatic urinary tract infection (UTI) [100], or existing UI may worsen during a UTI episode [101]. The evidence shows that the eradication of asymptomatic bacteriuria does not change the rate and severity of UI, hence it should not be treated as an effort to improve continence [102].

#### 3.3.5. Urethrocystoscopy

Urethrocystoscopy can evaluate urethra, sphincter and vesicourethral anastomosis, exclude foreign bodies such as clips or stones and identify suspicious bladder lesions [73]. Under an endoscopic view, the repositioning test can be assessed to recognize the best candidate for sling insertion [103,104,105]. A positive repositioning test is strongly correlated with a successful outcome after male sling insertion [105].

#### 3.3.6. Urodynamics

Urodynamics help the physician to identify the factors that contribute to UI, obtain information about all aspects of LUT function, allow prediction of the treatment outcome and understand the reasons for previous treatment failures. Urodynamics involve the filling cystometry and the pressure-flow study. The specific tests of urethral function include urethral profilometry, Valsalva and detrusor leak point pressures, retrograde urethral resistance, etc. Urodynamics can be combined with electromyography to assess pelvic floor muscle activity or imaging (ultrasound or fluoroscopy). Typical post-radical prostatectomy urodynamics findings are detrusor overactivity (2–63%), detrusor underactivity (29–61%) and poor compliance (5–25%) [106]. Of these, 21% of overactivity and 47% of underactivity cases are considered de novo. The urodynamics parameters that predict a poor treatment outcome are a low maximum cystometric capacity, poor compliance, high-amplitude DO, a low bladder contractility index, a high bladder outlet obstruction index and a low vesical leak point pressure [106]. The recommendation to perform urodynamics for male UI, considering the invasive treatment, is weak.

### 3.4. Conservative Treatment

In men with PPI, lifestyle interventions may include the management of obesity, smoking and physical activity, as well as diet. Modification of these aspects can have a positive impact on incontinence; however, the evidence on which these recommendations are based are studies with a predominantly female population [107].

With respect to containment, absorbent pads, urinary catheters, collection devices and penile clamps can be considered. One RCT compared a sheath drainage system, body-worn urinal (BWU), penile clamp and pads in a population of men with persistent urinary continence following treatment of prostate cancer. The study found that, with respect to quality of life and acceptability, pads and devices have different strengths, which make them particularly suited to certain circumstances and activities. According to the RCT, most men seem to prefer to use pads at night but would choose a mixture of pads and devices during the day [108].

Pelvic floor muscle training (PFMT) has been investigated thoroughly in reducing PPI. A Cochrane systematic review showed that there was no evidence from eight trials that PFMT, with or without biofeedback, is superior to the control in the reduction of PPI ≤ 12 months [109]. In contrast, one systematic review and meta-analysis showed that PFMT, with or without biofeedback and/or electrical stimulation, was effective for PPI, reducing the time to continence recovery [110]. In addition, one meta-analysis showed that guided programs with biofeedback or electrical stimulation are superior to PFMT alone in reducing PPI at one and three months [111]. In men undergoing robot-assisted radical prostatectomy (RARP), one systematic review showed that preoperative PFMT significantly reduces the duration and severity of early PPI [112]. Furthermore, a systematic review suggests that PFMT shortens the time to continence recovery after RARP [113]. With respect to the necessity of supervised PFMT, one RCT showed that written instructions alone show similar improvements in time to regain continence to supervised PFMT [114].

### 3.5. Medical Treatment

#### 3.5.1. Duloxetine

Table 1 reports medical treatment options for PPI.

Duloxetine is the most exhaustively studied drug for postprostatectomy stress urinary incontinence (SUI). Duloxetine is a selective serotonin and noradrenaline reuptake inhibitor which has an effect on Onuf’s nucleus in the spinal cord. By stimulating the pudendal nerve, the tension of the urethral sphincter is increased, and the detrusor muscle is relaxed [1]. Duloxetine has shown an effect in female SUI and has been licensed for this indication in some countries [115].

A recent systematic review analyzed the present evidence on duloxetine in postprostatectomy SUI [116]. The authors included eight studies, of which one was randomized noncontrolled [117]; two were randomized controlled trials [118,119]; four were prospective cohorts [120,121,122,123]; and one was a retrospective cohort [124]. The investigated cohorts differed, with some studies only including men with postprostatectomy SUI, while other studies also included a minor fraction of men with SIU following transurethral surgery such as TURP. Also, the dosing regimen of duloxetine differed and ranged from 30 mg to 60 mg once daily to 20 mg twice daily to 40 mg twice daily [116]. With respect to efficacy, the use of duloxetine following catheter removal after radical prostatectomy (RP), in combination with pelvic floor muscle training (PFMT), was compared with PFMT alone in two randomized trials with a follow-up of 4 to 9 months [117,118]. Both studies reported an earlier return of continence in those treated with duloxetine, with dry rates of 78% (compared with 52%) at 4 months and 96.5% (compared with 87%) at 1 year [117,118]. The only prospective, randomized, double-blind study comparing duloxetine with a placebo included 31 men with PPI, of whom 16 were treated with duloxetine [119]. At inclusion, both patient groups had an incontinence episode frequency (IEF) of 2–4 per day. The percentage of reduced IEF was significantly higher in the duloxetine group compared with the placebo group. No significant reduction in a 1h pad test was observed. In a recent systematic review, the dry rate, which was also defined variably between the studies as completely dry, as well as the use of one security pad, ranged from 25–89%, with a mean dry rate of 58%. The improvement in the mean pad number ranged from 12–100%, with a mean improvement of 61%, and the overall satisfaction was found to be 64% (50–77%) [116]. The most commonly reported side effects of duloxetine in the studies among men with PPI were fatigue, dry mouth, nausea, dizziness and constipation. The mean discontinuation rate due to adverse events was 21%, and the overall discontinuation rate of 38% [116]. In summary, duloxetine shows efficacy in the short-term cure or improvement of PPI and reduces the time to regain continence. Side effects are of considerable frequency, and a non-negligible proportion of men discontinue treatment due to side effects. In addition, the overall certainty of evidence from the existing studies seems moderate to low, with heterogeneity and relevant limitations in current studies. Hence, duloxetine is considered for off-label use for PPI in many countries, including the European Union.

#### 3.5.2. Antimuscarinic Drugs

While SUI is the most common form of incontinence following RP, studies have shown that up to 37.8% of male patients after RP are affected by de novo overactive bladder symptoms (OAB) [116]. Although the exact mechanism of OAB after RP is unclear, various factors such as detrusor overactivity, urethrogenic mechanisms and a defunctionalized bladder or bladder outlet obstruction have been suggested [125]. With respect to the pharmacological treatment of OAB following RP, antimuscarinic drugs have been investigated. Two prospective studies evaluated the effect of solifenacin (5 mg) once daily on the IPSS sub-score. While one study did not detect improvement in any storage LUTS at 3 months [126], another study showed significant improvement in urgency, frequency and nocturia at 3 and 6 months [127]. One randomized controlled trial including 27 patients found a significantly greater decrease in urgency, as assessed by the IPSS sub-score, with tolterodine (2 mg) than with no treatment within 30 days after catheter removal following RP [128]. The effect of solifenacin on continence recovery was further assessed in two RCTs, which did not specifically include the population with OAB [129,130]. One study compared midodrine against midodrine and solifenacin. At 4 months, the rate of continence defined as being pad free did not differ between the groups. However, the mean weight of the daily pads worn was significantly lower in the group of patients treated with the solifenacin. Also, the maximal cystometric capacity only increased after treatment with solifenacin [130]. Another randomized trial compared solifenacin to a placebo. The primary endpoint was the time from the first dose of the study drug to the date of urinary continence (defined as pad-free or dry security pad). The time to continence did not differ between both groups, whereas continence by the end of study and change in pads per day from the baseline was in favor of the solifenacin group [130]. In summary, there is some evidence that supports the use of antimuscarinic drugs in men suffering from OAB following RP.

#### 3.5.3. Beta-3 Agonists

Although international guidelines recommend antimuscarinics (AMs) or beta-3 agonists (B3As) as the first-line drug options, in clinical practice, AMs remain the first-line pharmacological treatment for OAB, while B3As are generally offered as a second-line treatment (De Nunzio et al. 2021). Tubaro et al. 2017 [131,132,133] focused on the male population of five phase III RCTs evaluating MIRA (50 mg). Overall, 1187 patients were included in the full analysis. Mirabegron (50 mg) was associated with a greater reduction in daily micturition versus a placebo in the pooled analysis, with a significant difference in favor of mirabegron versus the placebo of −0.37 (95% confidence interval [CI]: −0.74, −0.01, *p* < 0.05). Several studies have also suggested that mirabegron (50 mg) is well tolerated in men. In particular, serious adverse events and drug-related adverse events leading to drug discontinuation were uncommon (0.3% and 1.8%, respectively). The results of the efficacy and safety of vibegron (a new beta-3 agonist) have already been published, although the proportion of men in these mixed populations is low (6–14.5%) (De Nunzio et al. 2012) [131]. Notwithstanding this evidence, beta3 agonists are not included in the pharmacological armamentarium for the management of urinary incontinence after radical prostatectomy but can play a role only in the presence of a concomitant overactive bladder.

**Table 1 jcm-12-01190-t001:** Mechanism of action and cure rate of medical treatment options for PPI.

Options	Mechanism of Action	Cure Rate
**Duloxetine**	Stimulation of pudendal nerve, leading to increased tension of the urethral sphincter and relaxation of detrusor muscle	Dry rate: 25–89% [116]
**Antimuscarinic**	Antagonizing effect of muscarinic receptor subtypes in the bladder such as bladder contractions	Improvement in urge incontinence: up to 52% [129]

*Solifenacin*

*Tolterodine*	Decrease in early urge incontinence after RPRP: up to 53% [128]
**Beta-3** **Agonists** *Mirabegron*	Stimulation of beta-3 adrenoceptors in smooth muscle cells of detrusor induces detrusor relaxation	Reduced daily micturition: 37% [132]

### 3.6. Surgical Treatment

Surgery is the mainstay of male stress urinary incontinence management, especially in the case of sphincter deficiency, which mostly happens after prostatic surgery (radical prostatectomy or BPO relief surgery), or can be due to neurogenic bladder/sphincter dysfunction [7]. The available surgical options are the peri-urethral injection of bulking agents, male slings (autologous, fixed synthetic or adjustable), peri-urethral compression devices (e.g., Pro-ACT balloons), or circumferential compression devices (namely, artificial urinary sphincter (AUS)) implantation (Table 2).

For all these techniques, variable levels of evidence exist [7]. Among the recommended options, the respective indication for each treatment is based on the patient’s characteristics (degree of incontinence, history of pelvic radiation therapy, etc.), availability of the device, surgeon’s expertise and patient preferences. Surgical management of SUI is hardly ever proposed within 12 months after prostate surgery; however, some groups have proposed that treatment may be considered after 6 months in case of massive SUI [134].

#### 3.6.1. Bulking Agents

Bulking agents were proposed more than a decade ago for the management of mild to moderate SUI symptoms in men [135]. The principle of the treatment is to inject a bulking agent in the urethral wall, around the anatomic site of the striated sphincter, to increase the passive resistance of the urethra. Despite the fact that they have different characteristics, no bulking agent seems superior to another. The available evidence globally shows that the rate of cure is rather low (around 30%) [136] in selected cases. In the case of success, there is a high recurrence rate [135]. The adverse events are mild [136]. For these reasons, most clinical guidelines do not recommend the use of bulking agents for male SUI management [7,134].

#### 3.6.2. Male Slings

Different generations of male slings have been proposed in the past decades to treat SUI [137]. Autologous male sling procedures have been progressively abandoned, except in some specific cases (mostly neurogenic). Bone-anchored synthetic slings have also been stopped because of the high complication rate. Currently, the available slings are either fixed synthetic (mostly transobturator) or adjustable slings. Those devices have gained so much popularity that they are the most common procedure for SUI management in the United States (around 50% of cases) [138]. Many types, variations and brands are marketed, but all the devices are not supported by the same level of evidence. There is no adequate RCT comparing different types of slings.

The most widely studied device is the AdVance XP (Boston Scientific, Marlborough, MA, USA). The sling is placed through a transperineal approach via the transobturator route. An adequate dissection of the urethra is necessary to obtain an appropriate relocation and compression of the bulbar urethra during the sling tensioning. The efficacy has been established by numerous prospective studies, including long-term follow-up, after up to 5 years [139]. The cure rate, mostly defined as no pad use or a security pad, is around 50%, with a slight decline in the long term [137]. Few severe complications have been reported. Infection and erosion are uncommon after the procedure. The most frequent adverse events include transient urinary retention, post-operative pain and wound dehiscence. The fixed transobturator sling was more recently assessed in a randomized controlled trial against an AUS [140]. In this non-inferiority trial, the fixed transobturator male sling was noninferior to an AUS, with fewer adverse events. However, an AUS was associated with better subjective and quality of life outcomes in patients with severe SUI at the baseline. Nevertheless, there is growing evidence that the continence outcomes of fixed slings are significantly inferior to those of an AUS in patients with a moderate degree of post-prostatectomy SUI [141,142].

Other fixed male slings have been studied and marketed. The Virtue (Coloplast, Humlebaek, Denmark) quadratic sling is a 4-arm, large mesh implanted via the perineal route, with two transobturator arms and two prepubic arms, allowing for a large compression of the urethra. The cure rate, which seems impacted by the incontinence severity at the baseline, is estimated to be around 50% and declines with time. The reported adverse events were pain, acute urinary retention, infections or failure requiring explanting of the sling [143]. Other fixed male slings exist (e.g., IStop TOMS [144], with similar outcomes but are less popular and supported by a lower level of evidence.

Adjustable slings have been proposed with the intention to adjust the level of compression of the urethra, while still being minimally invasive. Those devices differ by their design, their adjustability mechanisms and route of implantation. The most widely studied devices include the Argus classic and ArgusT (Promedon, Cordoba, Argentina), ATOMS (A.M.I., Feldkirch, Austria) and Remeex (Neomedic International, Barcelona, Spain). A recent systematic review [145] reported cure rates of 69% after ATOMS implantation, and a 5.5% explantation rate. The results seemed inferior with the Remeex, with 53% success and 19% explantation. The Argus and ArgusT devices have been less studied, with a short-term success of about 70% but declining over time. Their explanation rate was reported around 10%. Overall, there is limited evidence regarding the efficacy of the adjustable male slings, and some evidence that the adjustable slings are associated with more complications [146,147]. As a result, those slings do not receive a positive recommendation in the current clinical guidelines.

One of the most challenging issues in male sling surgery is patient selection. Many predictive factors of failure and risk factors for complications have been proposed in the literature, including severe incontinence, previous urethral surgery, history of radiation therapy, the persistence of a residual sphincter function and detrusor dysfunction [147,148]. In those cases, the patient information may include a warning about a higher risk of failure.

#### 3.6.3. Peri-Urethral Balloons

The Pro-ACT (Uromedica, Inc., Plymouth, MN, USA) balloons are implanted by perineal approach, percutaneously, laterally under the bladder neck. A subcutaneous titanium port allows the adjustability of the volume of the balloons and thus the degree of peri-urethral compression. A number of prospective studies have shown the efficacy of Pro-ACT implantation for male SUI management [145]. The efficacy, mostly defined as the use of 0 or 1 pad post-operatively, is estimated to be 55% [30% to 75%]. The success rates are negatively influenced by the severity of incontinence and a history of radiation therapy. The latter has also been linked to more frequent complications and is thus considered as a contra-indication by some authors [6,147,148].

Pro-ACT balloon use has also been reported as a salvage therapy for mild, persistent or recurrent SUI symptoms after implantation of a fixed male sling [149,150].

Due to the very specific surgical technique, the complexity of indications, the need for a close follow-up, and the potentially high rate of erosion, mechanical failure and re-operations (estimated 25% explantation rate), Pro-ACT implantation should be considered only in specialized centers [6].

#### 3.6.4. Artificial Urinary Sphincter

The AUS is among the oldest surgical treatment of SUI, and expert surgeons specialized in incontinence all around the world have collectively gained enormous clinical experience regarding its indications, risks and benefits, including in the very long term. The principle of AUS is a circumferential compression of the urethra by a cuff, filled with liquid. The liquid can be pumped by digital compression of the pump to the pressure-regulating balloon, authorizing micturition. The original three-piece device (AMS800, Boston Scientific, Marlborough, MA, USA) is still the reference, although recent technical evolutions have been proposed.

The efficacy of AUS implantation has been established by a wide range of studies, including through a recent RCT [140]. An AUS seems particularly useful in moderate to severe incontinence, with a known high satisfaction rate and maintained success in the long term, despite up to 50% of patients possibly not being totally dry [151]. However, severe complications may occur: the risk of infection/erosion and mechanical dysfunction are evaluated as 8.5% [3.3–27.8] and 6.2% [2–13.8], respectively [150]. The risk of late reoperation is high, with an all-cause revision rate of around 50% of patients after 10 years of follow-up [152,153]. The risk/benefit ratio is then of utmost importance when discussing AUS implantation, which should be avoided in very frail patients with a high operative risk, inability to manipulate the pump or cognitive impairment.

The risk factors for the failure and complications of AUS are still debated among implanters. Radiation therapy has been the most frequent factor associated with failure, followed by previous anti-incontinence surgery (e.g., sling [154]), previous AUS implantation and/or explantation and frail urethra. Surgical tips and tricks have been developed through decades in reference and tertiary care centers, which are the ideal environment for AUS surgery [155].

Recently, some technical innovations have been presented with a remote control pump and pressure control system in preclinical studies [156]. Human experimentation is still pending.

**Table 2 jcm-12-01190-t002:** Mechanism of action and cure rate of surgical treatment options for PPI.

Options	Mechanism of Action	Cure Rate*(Dry Rate)*
**Peri-Urethral Injection** **of Bulking Agents**	Increase passive urethral resistance	<30% [135]
**SLING** **Non Adjustable** *AdVance XP™* *The Virtue_™_* *IStop TOMS_™_* **Adjustable** *ATOMS_™_* *Remeex_™_* *Argus™ and ArgusT_™_*	Relocation and compression of the bulbar urethra with the chance to adjust or not the level of compression on the urethra (adjustable and fixed respectively)	
Up to 70% [104,105,106,107,108,109,110,111,112,113,114,115,116,117,118,119,120,121,122,123,124,125,126,127,128,129,130,131,132,133,134,135,136,137,138,139]
Up to 50% [143]
Up to 50% [144]

Up to 69% [145,146,147,148]Up to 53% [145]Up to 70% [146,147]
**PERI URETHRAL BALLOONS** *Pro-ACT^™^*	Adjustable peri-urethral compression	
Up to 55% [150]
**ARTIFICICAL URINARY SPHINCTER** *AMS800^™^*	Circumferential compression of the urethra by a cuff filled with liquid (placed at the level of the bulbar urethra or with transcorporal approach	
Up to 90% [152,153,154]

## 4. Conclusions

In conclusion, post-prostatectomy UI depends on the patient’s biological and surgical factors and requires a careful assessment. The off-label use of duloxetine may lead to a short-term improvement in postprostatectomy SUI symptoms and QoL improvement, but a significant proportion of men will discontinue treatment due to side effects. Invasive therapies for the management of post-prostatectomy SUI include non-adjustable trans-obturator slings in men with mild-to-moderate incontinence and artificial urinary sphincters in men with moderate-to-severe incontinence.

## Figures and Tables

**Figure 1 jcm-12-01190-f001:**
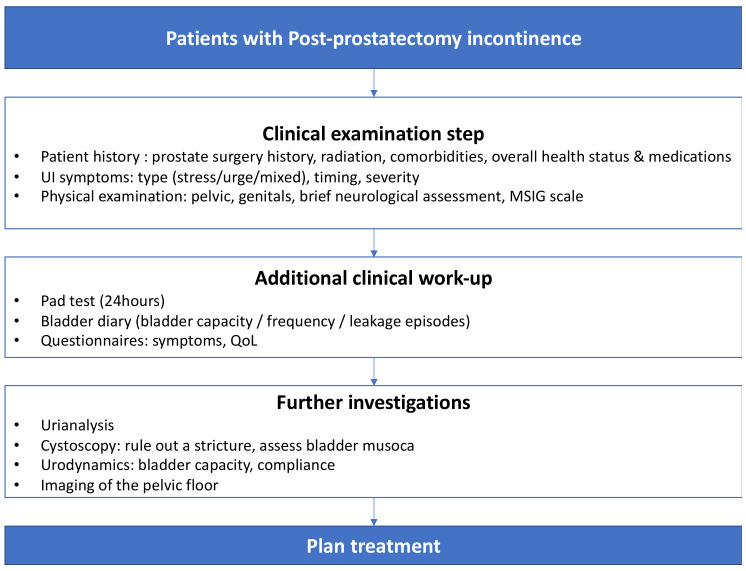
Diagnostic flowchart for patients with PPI3.3.

## Data Availability

Not applicable.

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
