# Peer review of "Latest Evidence on Post-Prostatectomy Urinary Incontinence"

_jcm, 2023, doi:10.3390/jcm12031190_

Round 1

Reviewer 1 Report

The introduction is too short for a review.

The same is true of the methodology. It is a review, but I do not have enough information about the methodology used to do it. We do not know which descriptors were used, which databases, etc. For that reason the review lacks methodological value since it may be a biased review. 

The results are not presented with tables, there is hardly any numerical data. 

In the medical sciences, numerical data are very important and make a difference.

This review, in my opinion, is an approximation to a scientific review, the methodology applied is not clear and therefore we cannot guarantee that the results are objective.

Author Response

The introduction is too short for a review.

RE: Thank you for your comments. Now the Introduction has been extensively re-edited.

The same is true of the methodology. It is a review, but I do not have enough information about the methodology used to do it. We do not know which descriptors were used, which databases, etc. For that reason the review lacks methodological value since it may be a biased review. 

RE: Based on your request, the following paragraph has been added to the manuscript:

The full details of the search strategy are available on the following website: https://d56bochluxqnz.cloudfront.net/documents/guideline-appendices/management-of-non-neurogenic-male-luts/Search-strategies-Non-Neurogenic-Male-LUTS-guideline-2022.pdf. In particular, from a total of 1,054 unique records, 663 were excluded (due to: being female, animal, child, duplicates, protocols, abstracts or paper on neurogenic UI). From the remaining 391 papers on male UI evaluated by our panel, 154 were included in the present review.

The results are not presented with tables, there is hardly any numerical data. In the medical sciences, numerical data are very important and make a difference.

RE: Now we have added 1 figure and 2 tables, with relative references.

This review, in my opinion, is an approximation to a scientific review, the methodology applied is not clear and therefore we cannot guarantee that the results are objective.

RE: I hope that now the methodology applied is clear, and the results now can be considered objective.

Reviewer 2 Report

The manuscript by Gacci et.al. provided a comprehensive and up-to-date summary on post RP urinary incontinence from pathophysiology to medical and surgical management. It is thoroughly researched, well written and easy to follow. 

One minor comment:

In the section of "surgical factors", please consider including literature evidence and discussion on the "Rocco stitich" for posterior reconstruction in help preserving continence. 

Author Response

The manuscript by Gacci et.al. provided a comprehensive and up-to-date summary on post RP urinary incontinence from pathophysiology to medical and surgical management. It is thoroughly researched, well written and easy to follow. 

RE: Thank you for your appreciation.  

One minor comment:

In the section of "surgical factors", please consider including literature evidence and discussion on the "Rocco stitich" for posterior reconstruction in help preserving continence. 

RE: based on your comment, we have added: “the preservation or reconstruction of membranous urethral supporting structures seems to significantly improve post-operative continence by fixing the urethra in an anatomic position during enhancement of abdominal pressure (the so called “Rocco stitch”).”

Round 2

Reviewer 1 Report

I think that is not suitable for publication. 

Author Response

The authors performed a comprehensive and synthetic up-to-date review of the latest evidence on post-prostatectomy incontinence. They adequately addressed all reviewer's comments and improved their ms.

RE: thank you. 

1. please re-check the references (e.g., reference 132 is the same as ref.135);

RE: Now the references have been modified according to your comment. 

2. it is incorrect to set the "cure rate" (defined as no-pad or one safety pad) associated to fixed slings at around 70%, it actually about 50%, as also found in the meta-analysis by Meisterhofer et al. (Meisterhofer K, Herzog S, Strini KA, Sebastianelli L, Bauer R, Dalpiaz O. Male Slings for Postprostatectomy Incontinence: A Systematic Review and Meta-analysis. Eur Urol Focus. 2020 May 15;6(3):575-592. doi: 10.1016/j.euf.2019.01.008. Epub 2019 Feb 2. PMID: 30718160.)

RE: The percentage has been changed and the suggested reference added.

3. p.11, line 441. I would add that there is also growing evidence that continence outcomes of fixed slings are significantly inferior to those of AUS in patients with a moderate degree of post-prostatectomy SUI [Sacco E, Gandi C, Marino F, Totaro A, Di Gianfrancesco L, Palermo G, Pierconti F, Racioppi M, Bassi P. Artificial urinary sphincter significantly better than fixed sling for moderate post-prostatectomy stress urinary incontinence: a propensity score-matched study. BJU Int. 2021 Feb;127(2):229-237.   -  
Khouri RK Jr, Ortiz NM, Baumgarten AS, Ward EE, VanDyke ME, Hudak SJ, Morey AF. Artificial Urinary Sphincter Outperforms Sling for Moderate Male Stress Urinary Incontinence. Urology. 2020 Jul;141:168-172]

RE: this section has been added (including both references)

4. history of bladder neck contracture should be listed among risk factors of male sling.

RE: The history of bladder neck contracture has been added as risk factor.